# Transcriptome Analysis of the Fruit of Two Strawberry Cultivars “Sunnyberry” and “Kingsberry” That Show Different Susceptibility to *Botrytis cinerea* after Harvest

**DOI:** 10.3390/ijms22041518

**Published:** 2021-02-03

**Authors:** Kyuweon Lee, Jeong Gu Lee, Kyeonglim Min, Jeong Hee Choi, Sooyeon Lim, Eun Jin Lee

**Affiliations:** 1Department of Agriculture, Forestry and Bioresources, College of Agriculture and Life Sciences, Seoul National University, Seoul 08826, Korea; lkwjudy@snu.ac.kr (K.L.); tto12321@snu.ac.kr (J.G.L.); mkl0228@snu.ac.kr (K.M.); 2Korea Food Research Institute, Wanju-gun, Jeollabuk-do 55365, Korea; choijh@kfri.re.kr; 3National Institute of Horticultural and Herbal Science, Rural Development Administration, Wanju-gun 55365, Korea; sylim84@korea.kr; 4Research Institute of Agriculture and Life Sciences, Seoul National University, Seoul 08826, Korea

**Keywords:** cell wall, cuticle, gene expression, harvest, storage, strawberry fruit, transcriptome, wax

## Abstract

Gray mold (*Botrytis cinerea*) is a fungal plant pathogen causing postharvest decay in strawberry fruit. Here, we conducted a comparative transcriptome analysis to identify differences in gene expression between the immature-green (IG) and mature-red (MR) stages of the “Sunnyberry” (gray mold-resistant) and “Kingsberry” (gray mold susceptible) strawberry cultivars. Most of the genes involved in lignin and alkane-type wax biosynthesis were relatively upregulated in “Sunnyberry”. However, pathogenesis-related proteins encoding R- and antioxidant-related genes were comparatively upregulated in “Kingsberry”. Analysis of gene expression and physiological traits in the presence and absence of *B. cinerea* inoculation revealed that the defense response patterns significantly differed between IG and MR rather than the cultivars. “Kingsberry” showed higher antioxidant induction at IG and upregulated hemicellulose-strengthening and R genes at MR. Hence, “Sunnyberry” and “Kingsberry” differed mainly in terms of the expression levels of the genes forming cuticle, wax, and lignin and controlling the defense responses. These discrepancies might explain the relative difference between these strawberry cultivars in terms of their postharvest responses to *B. cinerea*.

## 1. Introduction

Strawberry (*Fragaria* × *ananassa* Duch.) is an economically important horticultural crop. The export value of strawberries in Korea increases annually. Strawberry production and exportation have risen more than threefold over the past two decades [1]. Overall strawberry production reached nine million tons in 2016 and over ten percent of the fruit is exported. However, postharvest strawberry fruit is perishable and susceptible to mechanical damage, fungal decay, and rapid metabolism. Severe deterioration of postharvest fruit quality and economic loss may be the consequences of small scratches and bruises during packaging and shipping [2,3]. Strawberry fruit readily senesces after harvest and has a short shelf life because it is highly sensitive to biotic and abiotic stress [4].

*Botrytis cinerea* is a devastating postharvest fungal pathogen on strawberry. It infects mainly the floral organs and is quiescent during fruit ripening. This necrotroph obtains its nutrients from dead fruit tissues [5]. When *B. cinerea* spores attach to the host surface, they extend germ tubes to reach suitable infection sites. They then penetrate the host cells via appressoria and infection hyphae [6]. *B*. *cinerea* secretes cutinase and polygalacturonase (PG) as well as phytotoxic compounds to facilitate host penetration and infection [7].

Plants have evolved defense mechanisms against postharvest fungal pathogens [8]. Cuticles and cell walls are primary barriers that limit pathogen attack and release signaling molecules that activate defense responses [9,10,11]. Pattern recognition receptors and immune signaling transduction systems transmit signals that induce defense-related genes and transcription factors [12,13]. Xiong et al. [14] reported that > 1500 defense-related genes are induced after *B. cinerea* inoculation in mature-red (MR) strawberry fruit. Complex crosstalk among jasmonic acid, salicylic acid, ethylene, and abscisic acid participates in defense responses by regulating defense-, ripening-, and senescence-related genes [12,15]. Reactive oxygen species (ROS) and enzymatic or non-enzymatic antioxidant systems may play important roles in defense responses. ROS are secondary messengers that induce phytohormone responses and defense-related genes. Nevertheless, excessive ROS damage the fruit [16].

Next-generation sequencing identifies factors that contribute to fungal pathogen resistance in horticultural crops. Several researchers have attempted to identify the differences between resistant and susceptible cultivars in terms of defense response [17,18,19]. Certain studies have compared the transcriptomes of resistant and sensitive cultivars in the absence of pathogen attack [20,21]. Other studies have compared defense responses between unripe and ripe fruits because *B. cinerea* is quiescent in green fruit [22,23,24].

Resistance to postharvest fungal decay in strawberries depends largely on the genetic background of the cultivar. Dong et al. [25] reported that the fungal decay rates of postharvest strawberry fruit significantly differed among genotypes. They were not correlated with total phenolic or soluble sugar content but were correlated with firmness. No strawberry cultivar is fully resistant to *B. cinerea.* Studies on various *B. cinerea*-resistant strawberry cultivars have been conducted. An earlier work assessed postharvest storability in five different strawberry cultivars developed in Korea [3]. The “Sunnyberry” and “Kingsberry” varieties were the most resistant and susceptible to postharvest fungal decay, respectively.

To the best of our knowledge, no prior research has compared the transcriptomes of strawberry cultivars with different responses to *B. cinerea*. Here, we compared the transcriptomes of the aforementioned “Sunnyberry” and “Kingsberry” strawberry cultivars. We characterized their differentially expressed genes (DEGs) involved in pre-formed and induced defense responses. We determined the relative expression levels of several genes by quantitative polymerase chain reaction (qPCR) analysis. We evaluated the physiological traits of these fruit varieties in the presence and absence of *B. cinerea* inoculation. This study will help clarify the various defense responses in two different strawberry cultivars and identify the genetic factors that probably contribute to *B. cinerea* resistance in this crop. Our findings will also provide a theoretical basis for selecting and breeding strawberry cultivars with enhanced postharvest fruit storability.

## 2. Results

### 2.1. “Sunnyberry” Cultivar Is More Resistant to B. cinerea Than “Kingsberry” Cultivar

We used two methods to compare the resistance of “Sunnyberry” and “Kingsberry” strawberry fruit to *B. cinerea* at the MR stage (Figure 1). First, we stored the fruit at the MR stage at 10 °C for 10 d without *B. cinerea* inoculation. Disease symptoms initially appeared in “Kingsberry” after 3 d and in “Sunnyberry” after 5 d. Disease incidence rapidly increased in “Kingsberry” after 5 d storage and reached 68% after 10 d but only 19% in “Sunnyberry” by the same time (Figure 1B). Second, we inoculated strawberry fruits with *B. cinerea* conidia and stored them at 25 °C for 10 d. By the end of this period, most of the “Sunnyberry” fruits presented with only small lesions and very few pathogen mycelia. In contrast, most of the “Kingsberry” fruits were very soft and bore numerous *B. cinerea* hyphae (Figure 1A). The disease index markedly increased 5-day post-inoculation (dpi) (Figure 1C).

### 2.2. Comparison of “Sunnyberry” and “Kingsberry” Transcriptomes

The transcriptomes of the two strawberry fruit cultivars at the immature-green (IG) and MR stages were compared. Fifty-five million reads per sample were obtained through RNA sequencing (RNA-Seq) (Appendix A). In “Sunnyberry”, 692 DEGs were relatively upregulated only at IG and 2487 DEGs were relatively upregulated only at MR (Figure 2A). For “Kingsberry”, 950 DEGs were relatively upregulated only at IG and 2963 DEGs were relatively upregulated only at MR (Figure 2B). The DEGs were subjected to functional analysis by gene ontology (GO) term annotation (Figure 2C,D). The top five GO categories were “cellular process”, “cell”, “cell part”, “catalytic activity”, and “metabolic process” at both IG and MR.

### 2.3. Lignin Biosynthesis-Related Genes Were Relatively Upregulated in B. cinerea-Resistant “Sunnyberry” Cultivar

Genes related to cell wall metabolism were differentially expressed between the “Sunnyberry” and “Kingsberry” cultivars (Appendix A). At IG, endoglucanase genes were upregulated in “Sunnyberry” compared with those in “Kingsberry”. At MR, COBRA-like protein genes were upregulated in “Kingsberry” relative to “Sunnyberry”. At both IG and MR, exo-PG and pectate lyase (PL) genes were upregulated in “Sunnyberry” compared with those in “Kingsberry”. At MR, polygalacturonase 1β-like protein 3 (PGL3) [26], pectin acetylesterase (PAE), and pectin methylesterase (PME) genes were upregulated in “Kingsberry” relative to “Sunnyberry”.

Here, we identified DEGs involved in lignin biosynthesis (Figure 3; Appendix A). At both IG and MR, caffeic acid *O*-methyltransferase, cinnamoyl-CoA reductase 1 (CCR1), and cinnamyl alcohol dehydrogenase (CAD) genes were upregulated in “Sunnyberry” compared with those in “Kingsberry”. In contrast, at IG and MR, caffeoyl-CoA *O*-methyltransferase genes were upregulated in “Kingsberry” relative to “Sunnyberry”. Certain *CAD* genes were annotated as putative mannitol dehydrogenases. Nevertheless, these genes may be functionally redundant and could be considered *CAD* [27]. Class III peroxidases (PODs) and laccases (LACs) polymerize monolignol units into lignin. At MR, most DEGs annotated as *POD* and *LAC* were upregulated in “Sunnyberry” compared with those in “Kingsberry”.

### 2.4. Alkane-Type Cuticle Biosynthesis-Related Genes Were Relatively Upregulated in B. cinerea-Resistant “Sunnyberry”

Cuticular wax is biosynthesized by fatty acid elongation, decarbonylation, and acyl reduction [28]. Here, we identified DEGs encoding long-chain acyl-CoA synthetase (LACS), 3-ketoacyl-CoA synthase, 3-ketoacyl-CoA reductase, and 3-hydroxyacyl-CoA dehydratase (HCD) (Figure 4; Appendix A). At IG and MR, *HCD* was comparatively upregulated in “Kingsberry”. However, there was no trend in the expression of any other DEGs involved in fatty acid elongation. Protein ECERIFERUM 1-like (CER1) genes mediate decarbonylation of very long-chain fatty acids (VLCFAs) and were comparatively upregulated in “Sunnyberry” at IG and MR. Wax ester synthase/diacylglycerol *O*-acyltransferase (WSD) genes mediate acyl reduction of VLCFAs and were relatively upregulated in “Kingsberry” at both stages.

### 2.5. R Genes Were Relatively Upregulated in B. cinerea-Susceptible “Kingsberry”

Certain pathogenesis-related (PR) genes were identified as DEGs (Appendix A). At MR, endochitinase (CHI) and glucan endo-1,3-β-glucosidase (βGLU) genes were upregulated in “Kingsberry” compared with those in “Sunnyberry”. CHI and βGLU degrade chitins and glucans in fungal cell walls [29]. Thaumatin-like proteins (TLPs) are PR-5 class proteins. They were upregulated in “Sunnyberry” relative to “Kingsberry” at MR. In wild peanut (*Arachis diogoi* Hoehne), *TLP* was upregulated after *Phaeoisariopsis personata* (late leaf spot fungus) inoculation. In vitro, recombinant TLPs presented with activity against several fungal pathogens, including *B. cinerea* [30].

At both IG and MR, most DEGs encoding RPM 1-like protein and RPP13-like protein 1 were relatively upregulated in “Kingsberry”. Several bioinformatically predicted resistance gene analogs [31] were also identified as DEGs here.

### 2.6. Antioxidant System-Related Genes Were Relatively Upregulated in “Kingsberry”

At IG and MR, the antioxidant peroxiredoxin (PRX) and superoxide dismutase were comparatively upregulated in “Kingsberry” (Appendix A). The thiol peroxidase PRX participates in antioxidant defense. PrxIIE and PrxQ remove ROS, such as NO, that accumulate in response to biotic stress [32]. The 2-hydroxyisoflavanone dehydratase and dihydroflavonol-4-reductase genes are involved in flavonoid and isoflavonoid biosynthesis. At MR, they were upregulated in “Kingsberry” relative to “Sunnyberry”. At IG and MR, two leucoanthocyanidin reductase (LAR) genes were upregulated in “Kingsberry” compared with those in “Sunnyberry”.

### 2.7. Changes in Gene Expression Following B. cinerea Inoculation Revealed Differences between “Sunnyberry” and “Kingsberry” in Terms of Defense Response

Eight DEGs were selected at the IG and MR stages of “Sunnyberry” and “Kingsberry” in the presence or absence of *B. cinerea* inoculation. Their relative expression levels were measured by qPCR (Figure 5). In total, 5 dpi fruits were used for IG and 10 dpi fruits were used for MR. MR fruits sampled later as “Sunnyberry” and “Kingsberry” differed in terms of disease index at 5 dpi (Figure 1C).

In both cultivars, *CAD* was upregulated after *B. cinerea* inoculation only at IG while cellulose synthase-like protein D3 (CSLD3) was upregulated after *B. cinerea* inoculation only at MR. At MR, *CCR1* was relatively upregulated in “Sunnyberry”. However, *B. cinerea* inoculation dramatically lowered CCR1 mRNA in both cultivars at MR.

In the absence of *B. cinerea* inoculation, *CER1* was upregulated in “Sunnyberry” compared with that in “Kingsberry” at IG and MR. The opposite was true after *B. cinerea* inoculation. *LACS4* was upregulated after *B. cinerea* inoculation only at MR. In the presence and absence of *B. cinerea* inoculation, *LACS4* had higher expression levels in “Kingsberry” than in “Sunnyberry” at MR.

The relative mRNA level of *βGLU* presented with very wide SD. Hence, it did not significantly differ between cultivars. In the presence and absence of *B. cinerea* inoculation, *TLP1b* was relatively upregulated in “Sunnyberry” but was significantly suppressed at MR by inoculation. *RPM1* was comparatively upregulated in “Kingsberry” at IG and MR but its relative mRNA level was substantially higher at MR. *RPM1* was upregulated in MR fruits after *B. cinerea* inoculation. “Kingsberry” showed higher relative *RPM1* mRNA levels after *B. cinerea* inoculation.

### 2.8. Physiological Traits Were Altered by B. cinerea Inoculation but Did Not Significantly Differ between Cultivars at MR

Hydrogen peroxide (H_2_O_2_) content, total phenolic content, and total antioxidant activity were determined in the presence and absence of *B. cinerea* inoculation (Figure 6). At IG, the H_2_O_2_ content was significantly higher in “Kingsberry” than in “Sunnyberry” at 5 dpi (Figure 6A). Total phenolic compound content and total antioxidant activity were relatively higher in “Kingsberry” in the presence and absence of *B. cinerea* inoculation (Figure 6B,C). Total antioxidant activity significantly increased in “Kingsberry” after *B. cinerea* inoculation. 

At MR, however, there were no significant differences in H_2_O_2_ content, total phenolic content, or total antioxidant activity between cultivars or inoculation treatments. In contrast, the lignin content significantly increased after *B. cinerea* inoculation in “Sunnyberry” at MR (Figure 6D). However, there were no significant differences between cultivars.

## 3. Discussion

In the present study, we evaluated the transcriptomic differences between a strawberry cultivar showing resistance to *B. cinerea* (“Sunnyberry”) and one that is susceptible to this fungal pathogen (“Kingsberry”). We also measured the relative expression levels of eight selected DEGs and compared the physiological traits of these strawberry varieties in the presence and absence of *B. cinerea* inoculation.

Cell wall-degrading enzymes soften and crack the fruit and facilitate *B. cinerea* penetration and invasion [33]. They affect strawberry fruit firmness and susceptibility to *B. cinerea*. *PG* and *PL* were relatively upregulated in “Sunnyberry” while *PGL3*, *PAE*, and *PME* were comparatively upregulated in “Kingsberry” (Appendix A). Different pectin-degrading enzymes may have been activated in each cultivar and had different relative influences on fruit firmness and *B. cinerea* susceptibility.

Cell wall biosynthesis enzymes reinforce plant cells against pathogen attack by depositing lignin, strengthening cross-linkages, and changing wall component ratios [9]. Lignin is a major component of the secondary cell wall. In strawberry fruit, it is located mainly in the achenes and the vascular bundles [34]. Here, *CAD, CCR*, and other lignin biosynthesis-related genes were upregulated in “Sunnyberry” compared with those in “Kingsberry” (Figure 3; Appendix A).

Lignin biosynthesis-related genes may contribute to fungal pathogen resistance. *POD* and *CAD* were expressed at higher levels in anthracnose-resistant than in anthracnose-susceptible tea plants [21]. In strawberry fruit, terpinen-4-ol treatment reduced the incidence of decay, upregulated phenylpropanoid pathway-related genes, and increased the lignin content [35]. Lignin biosynthesis-related genes also affect strawberry fruit firmness. *CAD* and *CCR* were the most differentially expressed genes in three strawberry cultivars and strongly influenced fruit firmness [36]. In this study, *CAD* and *CCR* were relatively upregulated in “Sunnyberry”, and this cultivar presented with firmer and more pathogen-resistant fruit. Hence, these genes merit further investigation.

*B. cinerea* inoculation caused *CAD* upregulation at IG and *CSLD3* upregulation at MR (Figure 5). The main defense response of strawberry fruit against *B. cinerea* at MR might be the reinforcement of the xyloglucan chains. Caño-Delgado et al. [37] suggested that cellulose and lignin compensate for each other to restore and maintain structural cell wall integrity. Strawberry fruit may differentially regulate cell wall composition against fungal attack depending on the developmental stage. Here, however, there were no significant differences between “Sunnyberry” and “Kingsberry” in terms of cell wall metabolism-related gene expression.

Cuticles minimize postharvest water loss, protect against pathogens, and provide mechanical support during development and postharvest storage [10]. *CER1* was upregulated in “Sunnyberry” compared with that in “Kingsberry” and the opposite was true for *WSD* (Figure 4; Appendix A). *CER1* may play a crucial role in fruit wax formation [38]. Strawberries produce more alkane- than alcohol-type wax [39]. *CER1* and *LACS4* were relatively upregulated in “Kingsberry” after *B. cinerea* inoculation (Figure 5). Alkan et al. [40] reported that cuticle reinforcement occurs mainly in response to appressorium attachment. The present study indicated that cuticle reinforcement after *B. cinerea* perception is stronger in “Kingsberry” than in “Sunnyberry”.

The effects of cuticle biosynthesis-related genes on *B. cinerea* resistance are unclear. Changes in cuticle composition and quantity during development differ among crops [41]. There have been few published reports on strawberry fruit cuticles and waxes. It was indirectly implied that the strawberry fruit cuticle participates in fungal pathogen resistance. Strawberries treated with ozone demonstrated relatively lower incidence of fungal decay and thicker and denser cuticles [42]. Strawberry fruits treated with pulsed light exhibited relatively reduced decay rates, enhanced firmness, and denser and more highly organized cuticles and cell wall structures [43]. CO_2_ treatment upregulated *CER1* in strawberries and diminished fungal decay [2].

Most PR protein and R genes were upregulated in “Kingsberry” compared with those in “Sunnyberry” (Appendix A). *TLP*s were expressed at higher levels in “Sunnyberry” than in “Kingsberry” at IG and MR. Nevertheless, the relative *TLP1b* mRNA decreased following *B. cinerea* inoculation (Figure 5). Thus, the involvement of *TLP1b* in the defense responses of MR-stage strawberry against *B. cinerea* is unknown.

*RPM1* was more strongly upregulated in “Kingsberry” than in “Sunnyberry” after *B. cinerea* inoculation (Figure 5). At MR, it is R rather than PR genes that participate in *B. cinerea* defense responses. This finding is consistent with that reported by Mehari et al. [24]. They indicated that R genes were relatively upregulated in ripe strawberry fruit while PR protein expression was relatively upregulated in unripe strawberry fruit after *B. cinerea* inoculation. They also suggested that R gene upregulation may confer *B. cinerea* susceptibility to ripe fruit because these genes induce hypersensitive responses and cell death and facilitate necrotroph infection. Though “Kingsberry” may be more effective than “Sunnyberry” at inducing defense responses against *B. cinerea*, the former is paradoxically more susceptible to this fungal pathogen than the latter.

Several antioxidant systems-related genes were relatively upregulated in “Kingsberry” (Appendix A). At IG and MR, two *LAR* genes were expressed at higher levels in “Kingsberry” than in “Sunnyberry”. In strawberry fruit, LAR produces proanthocyanidins that have antifungal activity in vitro. LAR concentration is positively correlated with *B. cinerea* growth inhibition [44]. LAR accumulated in immature strawberry fruits after *B. cinerea* inoculation [45].

As H_2_O_2_ is a type of ROS, its accumulation increases antioxidant activity. H_2_O_2_ is also a substrate of PODs that polymerize monolignol units to lignin. At IG, the H_2_O_2_ content and total antioxidant activity were significantly higher in “Kingsberry” than in “Sunnyberry” after *B. cinerea* inoculation (Figure 6A,C). The relatively higher H_2_O_2_ content in “Kingsberry” might explain its comparative increase in total antioxidant activity. Lignin content significantly increased only in “Sunnyberry” at MR (Figure 6D). The flavonoid and lignin biosynthesis pathways share the upstream phenylpropanoid pathway. Ring et al. [46] reported that silencing the chalcone synthase gene in the flavonoid pathway increased lignin content. Hence, the flavonoid and lignin biosynthesis pathways might be mutually antagonistic. At IG, the flavonoid pathway might be a more effective defense against *B. cinerea* than lignin deposition while the opposite may be true at MR. Therefore, increases in antioxidant activity and phenolic compounds may not contribute to *B. cinerea* resistance in mature strawberries.

Figure 7 summarizes the differences between “Sunnyberry” and “Kingsberry” strawberry fruit in terms of gene expression at IG and MR and in the presence and absence of *B. cinerea* inoculation. Genes in red font were expressed at relatively higher levels at IG, while genes in blue font were expressed at relatively higher levels at MR.

## 4. Materials and Methods

### 4.1. Plant Materials

“Sunnyberry” and “Kingsberry” strawberry plants were cultivated in a glasshouse in the Strawberry Research Institute at Nonsan, South Korea. In mid-September, strawberry seedlings were transplanted at a density of 9000 ha^−1^ into soil in a plastic greenhouse. The daytime temperature was 15–25 °C and the nighttime temperatures were >10 °C in the autumn and 6 °C in the winter. Fruit at the IG stage was harvested in mid-March 2020. Fruit at the MR stage was harvested in mid-April 2020. Fruit uniform in size and color was selected. IG fruit were green and 1.5–2.0 cm in diameter while MR fruit were red and 3.5–4.0 cm in diameter. The fruit were immediately transported to the laboratory after harvest.

### 4.2. B. cinerea Inoculation

*B. cinerea* strain AIP1 (2006-111-00001) was obtained from the Center for Fungal Genetic Resources at Seoul National University, Seoul, South Korea, and cultured on potato dextrose agar at 25 °C. A conidial suspension was prepared using a 3-week culture. The conidia were collected with 2 mL sterile distilled water and filtered through four layers of gauze. The concentration of the conidial suspension was evaluated using a hemocytometer (Superior, Marienfeld, Germany) with a DE/Axio Imager A1 microscope (Carl Zeiss, Oberkochen, Germany) and adjusted to 1 × 10^5^ mL^−1^.

IG and MR strawberry fruits were disinfected by immersion in 2% (*v*/*v*) sodium hypochlorite for 3 min, rinsed in sterile distilled water for 2 min, and dried for 2 h on a clean bench. Each fruit was inoculated by dropping 10 μL conidial suspension onto its receptacle. A total of fifteen fruits for each stage were used for the inoculation, and the inoculated fruits were then stored at 25 °C and 100% relative humidity n a plastic box sealed with Parafilm^®^ (Bemis Co., Neenah, WI, USA).

### 4.3. Disease Assessment

Fifteen MR fruits for each cultivar were stored at 10 °C for 10 d, and the disease incidence rate (%) was calculated by dividing the number of decayed fruits by the total number of fruits and multiplying the quotient by 100. The disease incidence rate was calculated twice in 2019 and 2020. 

Other MR fruits were inoculated with *B. cinerea* and stored at 10 °C for 10 d, and their disease indices were rated as follows: 0, no symptoms; 1, symptoms on 0–10% of the fruit surface; 2, symptoms on 10–25% of the fruit surface; 3, symptoms on 25–50% of the fruit surface; 4, symptoms on >50% of the fruit surface (Appendix A). The disease incidence and indices were determined by visual examination. Fruits were considered to be decayed if they presented with symptoms of fungal pathogenesis or if their tissues were very soft.

### 4.4. RNA Extraction and cDNA Synthesis

Fruit samples 0.1 g in weight were flash-frozen in liquid nitrogen and pulverized. Total RNA was extracted from the powders with a Ribospin Seed/Fruit total RNA isolation kit (GeneAll, Seoul, Korea) according to the manufacturer’s instructions. Total RNA was extracted from MR fruit inoculated with *B. cinerea* via a slightly modified version of the method described by Reid et al. [47]. Each 0.15 g fruit sample was flash-frozen in liquid nitrogen and pulverized. The powder was added to 950 μL pre-warmed (65 °C) extraction buffer (2% (*w/v*) CTAB, 2.5 M NaCl, 300 mM Tris HCl (pH 8.0), 25 mM EDTA, and 2% (*w/v*) PVP) and the suspension was shaken vigorously. The sample was incubated in a drying oven at 65 °C and shaken vigorously every 5 min. The suspension was extracted twice with equal volumes of 24:1 (*v*/*v*) chloroform:isoamyl alcohol and centrifuged at 3500× *g* and 4 °C for 15 min. The top layer was transferred to a new tube and centrifuged at 25,000× *g* and 4 °C for 20 min. The supernatant was mixed with 0.1 vol of 3 M NaOAc and 0.6 vol isopropanol and stored at −20 °C for 30 min. The sample tube was centrifuged at 3500× *g* and 4 °C for 30 min to collect the nucleic acid pellet. The pellet was dissolved in 1 mL Tris-EDTA and mixed with 0.3 vol of 10 M LiCl.

The RNA was precipitated overnight by incubation at 4 °C and collected by centrifugation at 20,000× *g* and 4 °C for 30 min. The pellet was washed with 70% (*v*/*v*) ethanol and centrifuged at 3500× *g* and 4 °C for 5 min. The supernatant was discarded, and the pellet was centrifuged at 3500× *g* and 4 °C for 1 min to evaporate any remaining ethanol. The pellet was dissolved in 50 μL sterile distilled water. All solvents except the extraction buffer were ice-cold. RNA purity was assessed by gel electrophoresis. The A260:A230 and A260:A280 ratios were measured in a microplate spectrophotometer (BioTek Epoch, Winooski, VT, USA).

The total extracted RNA was used for RNA-Seq and qPCR analyses. For qPCR, cDNA was synthesized from 500 ng total RNA with a ReverTra Ace qPCR RT kit (Toyobo Co. Ltd., Osaka, Japan) according to the manufacturer’s instructions.

### 4.5. RNA-Seq, Data Processing, and Functional Analysis

Total RNA from the IG and MR stages of “Sunnyberry” and “Kingsberry” strawberry fruit was used. RNA-Seq was performed at the National Instrumentation Centre for Environmental Management of Seoul National University (Seoul, Korea) on a HiSeq 2500 platform (Illumina, San Diego, CA, USA) using 151 bp paired-end reads. Low-quality (<Q20), adaptor, and barcode sequences were trimmed out with Trim Galore v. 0.4.4 using the parameter “—gzip–paired”. Cleaned reads were mapped to the *Fragaria* × *ananassa* octoploid “Camarosa” annotated genome [48]. To identify the DEGs, the transcript levels were calculated as transcripts per million (TPM) using CLC Genomics Workbench v. 11 (Qiagen, Hilden, Germany) and its default parameters, namely, match score = 1, mismatch cost = 2, insertion cost = 3, deletion cost = 3, length fraction = 0.5, and similarity fraction = 0.8. The DEGs were filtered according to the false discovery rate criteria *p* < 0.01 and |fold-change| > 2. GO functional analysis was conducted using Blast2GO v. 5.2 and the NCBI (National Center for Biotechnology Information) database [49].

### 4.6. qPCR Analysis

qPCR analysis was conducted as described by Min et al. [3]. The cDNA was diluted to 100 ng μL^−1^ and combined with 5 μL of 2 × Real-Time PCR Master Mix containing SYBR Green 1 (BioFACT, Daejeon, Korea). PCR primers (Appendix A) were designed with the Primer 3 Plus server (http://www.bioinformatics.nl/cgi-bin/primer3plus/primer3plus.cgi). For each reaction, relative gene expression was normalized to the expression of the reference gene elongation factor 1-alpha (maker-Fvb3-3-snap-gene-215.43) by the 2^−ΔΔCt^ method [50].

### 4.7. H_2_O_2_ Content

The H_2_O_2_ content was determined according to the method of Junglee et al. [51], with slight modifications. Each 0.15 g sample was flash-frozen in liquid nitrogen, pulverized, combined with 1 mL cold (4 °C) trichloroacetic acid (0.1% *w/v*), and incubated at 4 °C for 10 min. All samples were centrifuged at 12,000× *g* and 4 °C for 20 min. Each 0.5 mL supernatant was mixed with 0.5 mL of 1 M KI and 0.25 mL of 10 mM potassium phosphate buffer (pH 7.0). The mixtures were incubated in the dark at 22 °C for 20 min. Absorbance was measured at 390 nm using a microplate spectrophotometer (BioTek Epoch, Winooski, VT, USA). A standard curve was plotted using various H_2_O_2_ concentrations.

### 4.8. Total Phenolic Content and Total Antioxidant Activity

Each 0.2 g sample was flash-frozen in liquid nitrogen, pulverized, combined with 10 mL of 80% (*v*/*v*) methanol, vortexed for 15 s, sonicated for 20 min, and centrifuged at 3000× *g* and 22 °C for 20 min. Each supernatant was transferred to a new 15 mL tube, diluted with an equal volume of 80% (*v*/*v*) methanol, and used in the subsequent assays.

Total phenolic content was determined with Folin–Ciocalteu reagent [52]. Fifty microliters diluted solution and 50 μL Folin–Ciocalteu reagent were added to 450 μL distilled water. The mixture was vortexed briefly and incubated at 22 °C for 5 min. Then, 150 μL of 20% (*w/v*) Na_2_CO_3_ and 200 μL distilled water were added, and the mixture was incubated in the dark at 22 °C for 30 min. Absorbance was measured at 750 nm using a microplate spectrophotometer (BioTek Epoch, Winooski, VT, USA). A standard curve was plotted using various gallic acid dilutions.

Total antioxidant activity was determined by evaluating the ABTS^•+^ scavenging activity [53]. The ABTS^•+^ solution was diluted with distilled water to OD_734_ = 0.7. Then, 1.2 mL diluted ABTS^•+^ solution was mixed with 10 μL sample. The mixture was vortexed briefly and incubated in the dark at 22 °C for 15 min. Absorbance was measured at 734 nm using a microplate spectrophotometer (BioTek Epoch, Winooski, VT, USA). A standard curve was constructed using various Trolox dilutions.

### 4.9. Lignin Content

Lignin was quantified according to the method of Ring et al. [46]. Analyses were performed on 0.2 g samples flash-frozen in liquid nitrogen and pulverized rather than on 0.05 g lyophilized fruit powder. Data were reported in OD_280_ g FW^-1^.

### 4.10. Statistical Analysis

RNA-Seq and qPCR analyses were conducted using three biological replicates (n = 3). H_2_O_2_, total phenolic, and lignin content and total antioxidant activity were determined using four biological replicates (n = 4). Significant differences between treatment means were determined with a two-sample *t*-test in SPSS v. 25.0 (IBM Corp., Armonk, NY, USA).

## 5. Conclusions

In the present study, we investigated transcriptomic differences between two strawberry cultivars with varying degrees of resistance to *B. cinerea*. The DEGs between “Sunnyberry” and “Kingsberry” at IG and MR disclosed that “Sunnyberry” exhibited relatively higher expression levels of genes related to lignin and cuticle biosynthesis. In contrast, “Kingsberry” displayed higher expression levels of genes regulating PR, R, and antioxidant system-related proteins. Examination of induced defense responses revealed that the physiological traits and the expression levels of genes controlling cell wall metabolism, cuticle biosynthesis, and defense response varied with cultivar and developmental stage. Our study demonstrated that “Sunnyberry” and “Kingsberry” differed in terms of their pre-formed physical barriers and induced defense responses. However, their defense strategies were effective at MR and contributed to *B. cinerea* resistance.

## Figures and Tables

**Figure 1 ijms-22-01518-f001:**
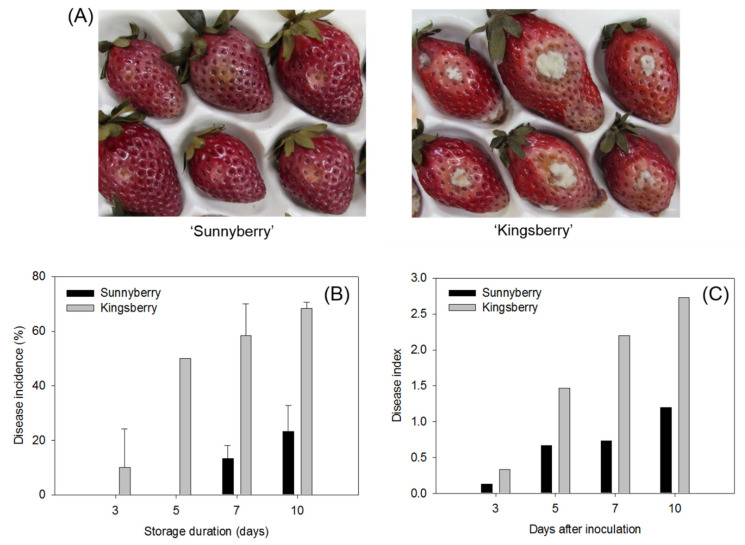
“Sunnyberry” and “Kingsberry” strawberry fruit inoculated with *Botrytis cinerea*. (**A**) Fruit at 10 d post-inoculation. (**B**) Disease incidence during 10 d storage at 10 °C in the absence of *B. cinerea* inoculation. (**C**) Disease index during 10 d storage at 25 °C post-inoculation. Data are means ± SD.

**Figure 2 ijms-22-01518-f002:**
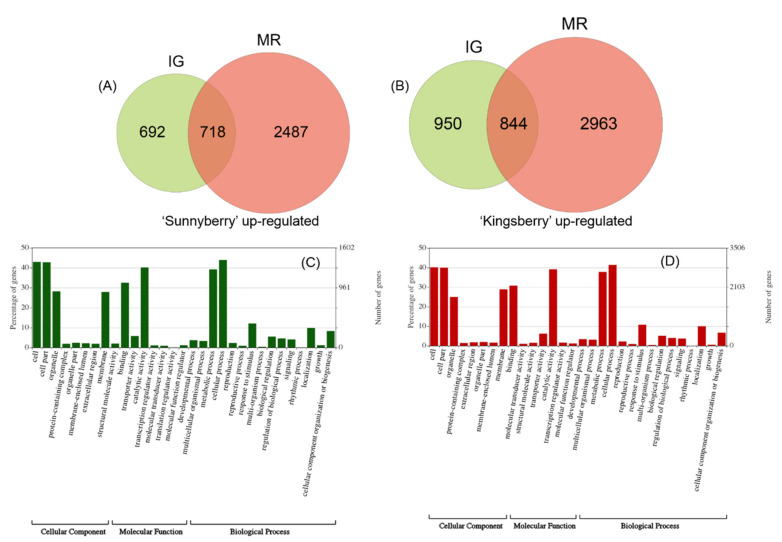
Transcriptomes of “Kingsberry” and “Sunnyberry” strawberry fruit. (**A**) Upregulated DEGs in “Sunnyberry” at IG and MR. (**B**) Upregulated DEGs in “Kingsberry” at IG and MR. GO annotation of DEGs at IG (**C**) and MR (**D**). DEGs were filtered using the false discovery rate (FDR) criteria *p* < 0.01 and |Fold change| > 2. DEGs, differentially expressed genes; GO, gene ontology; IG, immature-green; MR, mature-red.

**Figure 3 ijms-22-01518-f003:**
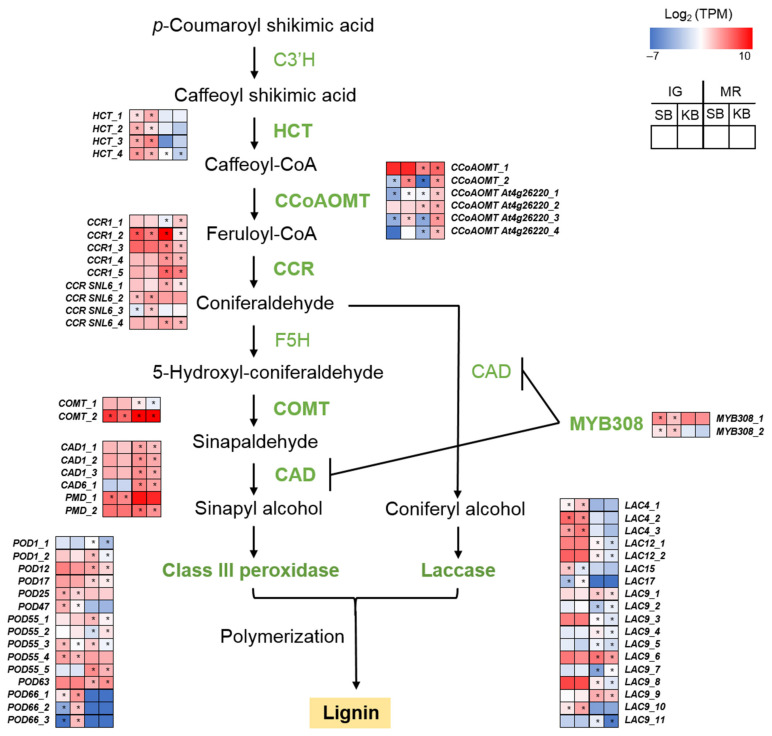
Lignin biosynthesis gene expression patterns at IG and MR of “Sunnyberry” and “Kingsberry” strawberry fruit. C3’H, *p*-coumaroyl shikimate 3’-hydroxylase; CAD, cinnamyl alcohol dehydrogenase; CCoAOMT, caffeoyl-CoA *O*-methyltransferase; CCR, cinnamoyl-CoA reductase; COMT, caffeic acid *O*-methyltransferase; F5H, ferulic acid 5-hydroxylase; HCT, shikimate *O*-hydroxycinnamoyltransferase; IG, immature-green; KB, “Kingsberry” cultivar; LAC, laccase; MR, mature-red; PMD, probable mannitol dehydrogenase; POD, peroxidase; SB, “Sunnyberry” cultivar; TPM, transcripts per million. Asterisks indicate statistically significant differences between cultivars (FDR *p* < 0.01; |Fold change| > 2).

**Figure 4 ijms-22-01518-f004:**
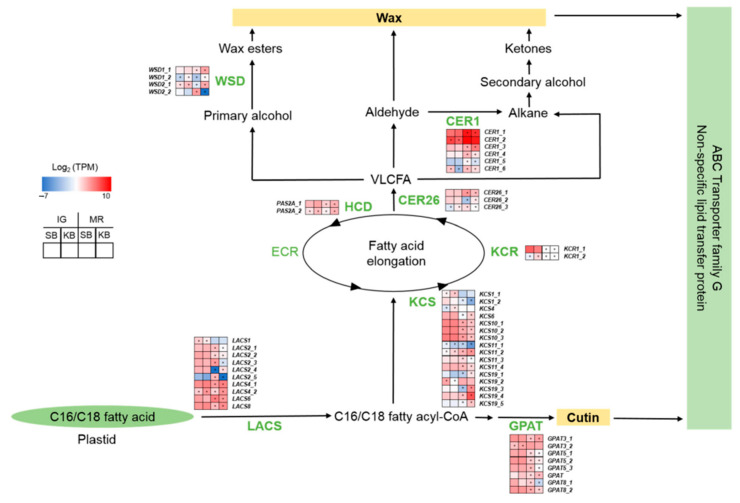
Expression patterns of cuticle biosynthesis genes at IG and MR in “Sunnyberry” and “Kingsberry” strawberry fruit. CER, protein ECERIFERUM; ECR, enoyl-CoA reductase; GPAT, glycerol-3-phosphate acyltransferase; HCD, 3-hydroxyacyl-CoA dehydratase; IG, immature-green; KB, “Kingsberry” cultivar; KCR, 3-ketoacyl-CoA reductase; KCS, 3-ketoacyl-CoA synthase; LACS, long-chain acyl-CoA synthetase; MR, mature-red; PAS2A, very-long-chain (3R)-3-hydroxyacyl-CoA dehydratase PASTICCINO 2A; SB, “Sunnyberry” cultivar; TPM, transcripts per million; WSD, wax ester synthase/diacylglycerol *O*-acyltransferase. Asterisks indicate statistically significant differences between cultivars (FDR *p* < 0.01; |Fold change| > 2).

**Figure 5 ijms-22-01518-f005:**
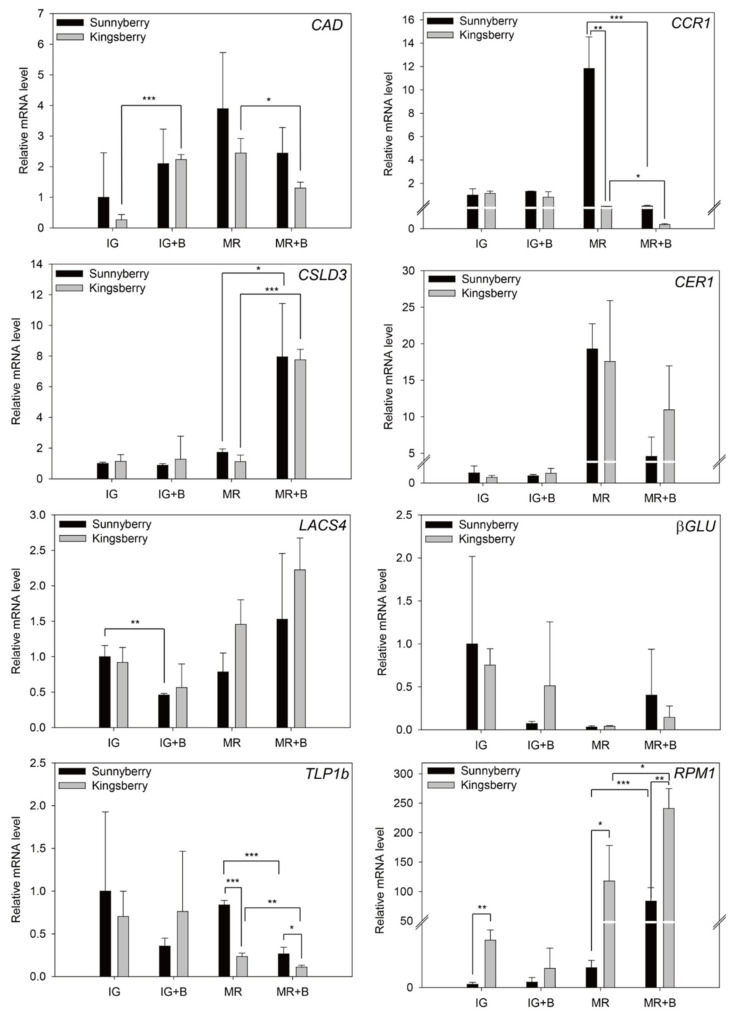
qPCR analysis of eight DEGs in “Sunnyberry” and “Kingsberry” strawberry fruit in the presence or absence of *Botrytis cinerea* inoculation. Relative mRNA levels of genes related to cell wall metabolism (*CAD*, *CCR1*, and *CSLD3*), cuticle biosynthesis (*CER1* and *LACS4*), and defense responses (*βGLU*, *TLP1b*, and *RPM1*). *βGLU*, glucan endo-1, 3-β-glucosidase; *CAD*, cinnamyl alcohol dehydrogenase; *CCR1*, cinnamoyl-CoA reductase 1; *CER1*, protein ECERIFERUM 1; *CSLD3*, cellulose synthase-like protein D3; IG, immature-green fruit without inoculation; IG + B, immature-green fruit at 5 days after inoculation; *LACS4*, long-chain acyl-CoA synthetase 4; MR, mature-red fruit without inoculation; MR + B, mature-red fruit at 10 days after inoculation; *RPM1*, disease resistance protein RPM 1-like; *TLP1b*, thaumatin-like proteins. Data are means ± SD for three biological replicates. Asterisks indicate statistically significant differences between cultivars. *, *p* < 0.05; **, *p* < 0.01; ***, *p* < 0.001.

**Figure 6 ijms-22-01518-f006:**
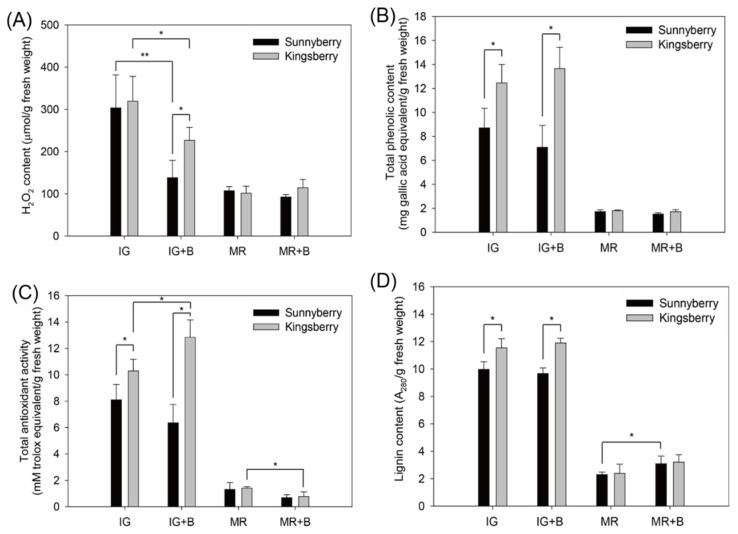
Changes in the physiological properties of “Sunnyberry” and “Kingsberry” strawberry. (**A**) H_2_O_2_ content, (**B**) total phenolic content, (**C**) total antioxidant activity, and (**D**) lignin content. IG, immature-green fruit without inoculation; IG + B, immature-green fruit at 5 d after inoculation; MR, mature-red fruit without inoculation; MR + B, mature-red fruit at 10 d after inoculation. Data are means ± SD for four biological replicates. Asterisks indicate statistically significant differences between cultivars. *, *p* < 0.05; **, *p* < 0.01.

**Figure 7 ijms-22-01518-f007:**
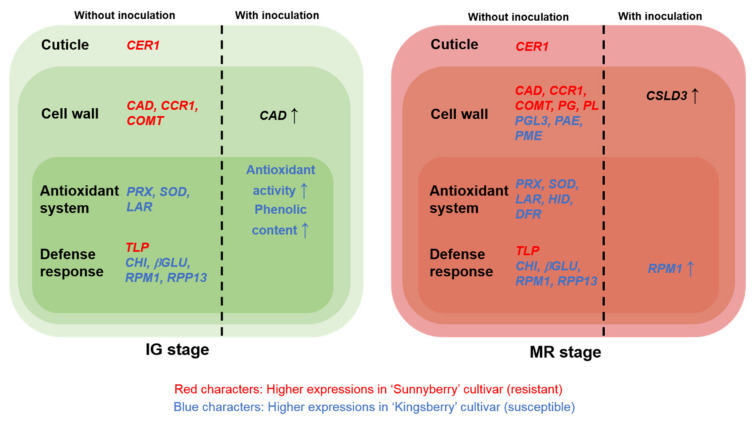
Differences between cultivars at IG and MR in terms of gene expression and defense responses against *Botrytis cinerea*. *βGLU*, glucan endo-1, 3-β-glucosidase; *CAD*, cinnamyl alcohol dehydrogenase; *CCR1*, cinnamoyl-CoA reductase 1; *CER1*, protein ECERIFERUM 1; *CHI*, chitinase; *COMT*, caffeic acid *O*-methyltransferase; *CSLD3*, cellulose synthase-like protein D3; *DFR*, dihydroflavonol 4-reductase; *HID*, 2-hydroxyisoflavanone dehydratase; IG, immature-green; *LAR*, leucoanthocyanidin reductase; MR, mature-red; *PAE*, pectin acetylesterase; *PG*, polygalacturonase; *PGL3*, polygalacturonase 1β-like protein 3; *PL*, pectate lyase; *PME*, pectin methylesterase; *PRX*, peroxiredoxin; *RPM1*, disease resistance protein RPM 1-like; *RPP13*, disease resistance protein RPP 13-like; *SOD*, superoxide dismutase; *TLP*, thaumatin-like protein.

## Data Availability

All data are available upon reasonable request.

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
