# Peer review of "Transcriptome Analysis of the Fruit of Two Strawberry Cultivars “Sunnyberry” and “Kingsberry” That Show Different Susceptibility to Botrytis cinerea after Harvest"

_ijms, 2021, doi:10.3390/ijms22041518_

Round 1

Reviewer 1 Report

The manuscript by Lee and coworkers deals with the assessment, at the transcriptomic level, of the response of two strawberry cultivars, namely ‘Sunnyberry’ and ‘Kingsberry’, to the post-harvest attack of B.cinerea.

The manuscript is well written and informative but unfortunately does not provide conclusive proofs about the actual plant defence response effectively involved in resistance against the fungal invasion.

I'd suggest to the authors to stress more explicitly -if this is the case- that most of the observed resistance may not be due to genes induced upon infection, but to the inherent features of the fruits, that develop during fruit growth (i.e cuticle structure, wax content and so on) independently of the fungal challenge.

Last but not least, a more comprehensive comparative study, taking into consideration various resistant cultivars, would provide much more insights into the actual genes/mechanisms/features involved in resistance, as it would allow identifying shared features, if any, which would strengthen the overall data (it would be somehow a "validation"). Such an approach would greatly improve the quality of the manuscript and its scientific soundness. Thus, I would encourage the authors to include such data, if available.

A manuscript including the aforementioned improvements would be much more suitable for publication on IJMS and of interest for a much wider readership.

Reviewer 2 Report

The transcriptome of strawberry fruits and the effects of Botrytis have been studied previously by others (e.g. https://doi.org/10.3389/fpls.2019.01131). However, the current papers expands the current knowledge and presents comparative data from a Brotytis resistant and susceptible cultivar. The results are interesting and clearly add new value to the research on this pathogen on strawberry with more options in the future for a better control. The study is well done, and well presented in the manuscript.

Round 2

Reviewer 1 Report

The authors replied to the reviewer's concerns by providing sufficient explanations.